# Temporal shifts in HIV-related risk factors among cohorts of adolescent girls and young women enrolled in DREAMS programming: evidence from Kenya, Malawi and Zambia

Sanyukta Mathur,[1] Craig J Heck [ORCID],[2] Sangram Kishor Patel,[3] Jerry Okal,[4] Effie Chipeta,[5] Victor Mwapasa,[5] Wanangwa Chimwaza,[5] Maurice Musheke,[6] Bidhubhusan Mahapatra,[3] Julie Pulerwitz,[1] Nanlesta Pilgrim[7]

For numbered affiliations see end of article.

**Correspondence to**
Dr Sanyukta Mathur; smathur@popcouncil.org

## ABSTRACT

**Objectives** To assess temporal shifts in HIV risk factors among adolescent girls (AG, aged 15–19 years) and young women (YW, aged 20–24 years) in Kenya, Malawi and Zambia.

**Design** Prospective cohorts with two time points (Kenya: 2016/2017, 2018; Malawi: 2017, 2018; Zambia: 2016/2017, 2018)

**Setting** Community-based programming.

**Participants** 1247 AG (Kenya: 389, Malawi: 371, Zambia: 487) and 1628 YW (Kenya: 347, Malawi: 883, Zambia: 398)

**Intervention** Determined, Resilient, Empowered, AIDS-free, Mentored and Safe (DREAMS), a multisectoral approach to reduce AGYW's HIV vulnerability by delivering a package of tailored, multilayered activities and services.

Primary and secondary outcome measures: HIV testing, sexually transmitted infection (STI) symptom experience, number of sexual partners, condom use (consistently, at last sex), transactional sex, experience of physical violence (from intimate partners) and sexual violence (from intimate partners and strangers/non-partners).

**Results** Changes in HIV-related risk behaviours among DREAMS participants varied by age group and country. Among AG, HIV testing increased (Kenya and Zambia) and sexual violence from partners (in Kenya and Malawi) and non-partners (in Malawi) decreased. Among YW, HIV testing increased and STI experience decreased in Malawi; consistent condom use decreased in Kenya; transactional sex increased in Kenya and Zambia; and physical violence (in Malawi) and sexual violence from partners (in Kenya and Malawi) and non-partners (all three countries) decreased over time.

**Conclusions** Improvements in HIV testing and reductions in experiences of sexual violence were coupled with variable shifts in HIV-related risk behaviours among DREAMS participants in Kenya, Malawi and Zambia. Additional consideration of AGYW's risk circumstances during key life transitions may be needed to address the risk heterogeneity among AG and YW across different contexts.

## Strengths and limitations of this study

► Longitudinal cohorts of adolescent girls and young women in three sub-Saharan countries were followed across two time points, over a time period of 12–16 months.

► Comprehensive quantitative survey captured knowledge, attitudes and HIV-related risk behaviours and exposure to a community-based multisectoral HIV prevention programme.

► Multivariate analyses examined change over time in factors associated with HIV acquisition.

► Due to a lack of a comparison group, our findings could have been confounded by unobserved changes or secular trends in the study sites and/or the ageing of the cohort.

## INTRODUCTION

Four decades into the HIV epidemic, adolescent girls and young women (AGYW, aged 15–24 years) remain at high risk for HIV. Globally, over 7000 AGYW seroconvert weekly, and even though they comprise 1/10th of the population, 20% of sub-Saharan Africa's (SSA) seroconversions occur among AGYW.[1] In eastern and southern Africa, which has the world's highest regional prevalence (7.0% (5.9%–7.9%)),[2] the HIV epidemic is magnified by gendered disparities. AGYW's prevalence and incidence rates in Kenya, Malawi and Zambia are double that (or more) of their male counterparts,[3–5] and when compared with all AGYW in SSA, all three countries have a prevalence that is equal to or greater than the regional average.[6] Though AG (aged 15–19 years) and YW (aged 20–24 years) are a priority population, the breadth, severity and context of their HIV risk is not homogeneous, for among females aged

15–29, HIV incidence is highest among AG in Kenya and YW in Malawi and Zambia.[6]

Underscoring these disproportionate HIV rates, a range of behavioural, biological, psychosocial and structural factors fuel AGYW's vulnerability to HIV.[6] For instance, AGYW who have multiple sex partners are at higher risk for HIV due to increased exposure,[7 8] and experience of sexually transmitted infection (STI) symptoms has been linked to risky behaviours and biological vulnerability.[7 9] Physical and sexual violence, which is known to cause physical and psychological trauma, is associated with AGYW's HIV risk when perpetrated by partners[10–13] and non-partners.[11 13] In addition, non-marital partnerships are associated with age–disparate relationships and inconsistent condom use among AGYW and sexual concurrency among their male partners.[14–16] Further, economic disenfranchisement and limited social capital push some AGYW to seek out sexual partners for financial support; these transactional arrangements often include relationships with older men[17 18] and increase AGYW's HIV risk.[19–21] Negative life and social transitions—such as experiencing paternal loss,[22–25] divorce/widowhood[7 26] and school dropout[7 18 27]—have also been associated with heighted engagement in risky sexual behaviours and precarious decision making.

AGYW's HIV risk factors are multifaceted and ecological, making them difficult to change using siloed, unicomponent interventions.[28] In response, the US President's Emergency Plan for AIDS Relief (PEPFAR), the Bill & Melinda Gates Foundation, Girl Effect, Johnson & Johnson, Gilead Sciences and ViiV Healthcare initiated the Determined, Resilient, Empowered, AIDS-free, Mentored and Safe (DREAMS) Partnership in 2015, with the goal to reduce HIV acquisition among AGYW in 14 sub-Saharan countries (including Kenya, Malawi, and Zambia) and Haiti. Using public and private delivery channels, DREAMS delivers a comprehensive package of evidence-based activities that aim to provide vulnerable AGYW with knowledge, life skills and access to social protection interventions (eg, educational support). DREAMS also aspires to create an enabling environment that supports AGYW's health and development by offering events and services that engage parents and the broader community.[29–32] However, there is limited evidence about how this AGYW-focused approach, which extends beyond behavioural or health sector interventions, is influencing HIV-related risk among AGYW.

To contribute to the growing body of literature around comprehensive multisectoral HIV prevention programmes, we assess temporal shifts in factors that have been previously linked to HIV-acquisition among DREAMS participants across three country contexts. Using an age-stratified approach to illuminate risk heterogeneities, we examine change over time in proximal behavioural, biological, and experiential HIV-related risk factors to assess if participation in the DREAMS programme positively influenced participants' vulnerability. This study provides initial insight on the change over time in HIV-related risk factors among cohorts of AG and YW DREAMS participants in Kenya, Malawi and Zambia, with the aim to inform the refinement or development of comprehensive multisectoral HIV prevention efforts.

## METHODS
### Program description
The DREAMS programme aims to empower AGYW and reduce their risk of HIV and gender-based violence by delivering a package of tailored, multilayered activities, services and interventions.[29–32] The DREAMS approach has been described in detail elsewhere.[29] Briefly, the core package of interventions includes four components: (1) empowering AGYW—including condom promotion, PrEP provision (only approved and provided in Kenya at the time of study), postviolence care, HIV testing services (HTS), increasing access to voluntary family planning services and social asset building (eg, knowledge and skills to increase self-efficacy); (2) strengthening families through parenting/caregiver programmes, education subsidies and combination socioeconomic approaches; (3) mobilising communities to address harmful community norms and practices, and school based HIV and violence prevention; and (4) reducing risk of AGYW's sexual partners by increasing access to HIV prevention and care services. DREAMS implementing partners (IP) were expected to reach one-half of all vulnerable AGYW in their study communities during the study implementation time period.[29] Country teams recruited AGYW based on a range of factors that constituted vulnerability (eg, being out-of-school, orphaned, history of violence experience, STI experience). The programme uses designated spaces within the community, where AGYW engage, often weekly, in the DREAMS activities. The safe spaces, which act as a programmatic hub and link AGYW peers and mentors, facilitated instruction on health and life skills, and referrals to youth-friendly health services and social protection interventions.

PEPFAR determined the target DREAMS programme areas, with input from local officials and other stakeholders, based on their high HIV prevalence and additional factors, such as high poverty levels, population density and/or population growth. In each site, IP used systematic selection processes to identify (eg, visiting and screening households in their community catchment area,[33] recruit and enrol eligible AGYW beneficiaries.[34–39] In Malawi, an additional eligibility criterion was out-of-school status. In each country context, a primary package of interventions[37 39 40] was delivered, with the expectation that all DREAMS participants would receive this primary package and additional interventions, depending on individual need or vulnerability.

### Study sites
In Kenya and Zambia, the sites for this study were purposely selected, in consultation with PEPFAR,

local stakeholders and IPs, to be representative of key geographical characteristics (eg, urban/periurban/rural) of DREAMS programme communities in each country. In Malawi, the study sites were selected from each of the two districts involved in DREAMS programming at the time. In total, the study was conducted in two sites in Kisumu County, Kenya; four sites in Malawi's Zomba and Machinga districts; and two sites in Lusaka and Ndola, Zambia.

## Study population

Two rounds of data were collected with study participants. Round 1 data were collected in 2016/2017. Eligible survey participants were females aged 15–24 years residing in the study catchment area who were enrolled in the DREAMS programme and intended to stay in the area for the subsequent year. At round 1, study participants were identified using an age-stratified random sample from the DREAMS programme beneficiary rosters (Kenya (n=5997), Malawi (proportionate to the differently sized districts, n=4738)) prepared by the programme IPs, whereas a census of all AGYW enrolled in DREAMS was conducted in Zambia. Potential participants were approached by study staff and invited to take part in the study. Using a conservative estimate (ie, 50%) of baseline prevalence for the primary outcomes (eg, HIV testing), the samples were powered at 80% to detect a minimal 10%–20% change. In Kenya, 474 AG and 440 YW were interviewed from October 2016 to February 2017. In Malawi, 530 AG and 1133 YW were interviewed from July 2017 to September 2017. In Zambia, 585 AG and 479 YW were interviewed from November 2016 to April 2017. Twenty respondents in Kenya, 33 in Zambia and 3 in Malawi refused to participate due to lack of parental consent, in ability to locate the participant or limited time availability/other events that prohibited participation.

A second round of data was collected with round 1 respondents in 2018 after approximately 12–16 months of exposure to DREAMS. At round 2, 80.5% of Kenya's respondents (389 AG and 347 YW) were reinterviewed from April to June 2018, 75.4% of Malawi's respondents (371 AG and 883 YW) were reinterviewed from September to November 2018, and 83.2% of Zambia's respondents (487 AG and 398 YW) were reinterviewed from April to May 2018. Overall, lost to follow-up occurred due to extended or permanent out-migration from the study communities, as confirmed by the programme team and through repeated attempts to recontact participants by the research team. Those not reinterviewed had relocated to towns outside study settings due to family, business or school reasons. Comparison of baseline characteristics and outcome values between respondents who were followed up and those who were lost show some differences by age group or country, but do not show any systematic patterns of loss (see online supplemental table 1). For instance, Kenyan AG who could not be reinterviewed were more likely to be out of school ($p \leq 0.05$) and

orphaned ($p \leq 0.01$), but this was not the case for YW in Kenya or AG and YW in Malawi and Zambia.

## Study procedures

Respondent consent (or parent/guardian consent and minor assent) was obtained before each round of data collection. Using a tablet-based survey instrument in the respondent's chosen language (Kiswahili and Luo in Kenya, Chichewa and Yao in Malawi, Bemba and Nyanja in Zambia, or English in all countries), female interviewers captured respondents' sociodemographic characteristics; programme engagement and exposure; and knowledge, attitudes and behaviours related to HIV/AIDS. Interviews were conducted in private yet convenient locations to the respondents (eg, room in respondent's home, nearby field or nearby community centre), and out of earshot of parents, guardians or other community members.

## Measures
### Cohorts' characteristics

The study cohorts only included respondents who provided information at both rounds 1 and 2. We explored the cohorts' sociodemographic characteristics—such age (15–19 vs 20–24) and marital (ever (currently, formerly) vs never married), schooling (in school vs out of school), and orphanhood (both parents alive vs at least one parent deceased) status, and using round 2 data only, we examined their participation in social asset building activities and exposure to (ie, being offered the service/activity by DREAMS staff) various DREAMS programme components—such as HTS, health services (condoms, PrEP, postviolence care, contraception, STI testing, pregnancy consultation), educational support (money for fees, uniforms, transport or other help with schooling expenses in Kenya and Zambia; back to school support in Malawi) and economic support (cash transfer in Kenya and Zambia, food support in Malawi). The study sites remained the same across the two rounds of data collection (Kenya: Nyalenda A, Kolwa East; Malawi: Zomba, Machinga; Zambia: Lusaka, Ndola).

We assessed engagement in DREAMS' primary intervention by asking respondents if they participated in the safe space groups (yes vs no) by round 2. Additionally, to examine the level of exposure to all DREAMS activities, we asked participants about the frequency and duration (short (<1week, 1week) vs long (1month, several months)) of their programme interruptions. Among those that experienced periods of non-participation, we used the median number of interruptions (three in all countries) to classify respondents as experiencing few (<median) or many (≥median) interruptions.

## Outcomes

We focused on factors that are known to facilitate or mitigate AGYW's HIV risk. HIV testing in the last 12 months and experiencing STI symptoms (genital warts, painful urination, vaginal discharge, genital ulcer) in the last 6 months were coded as binary (yes vs no) variables.

Among those who had sexually debuted, we truncated respondents' number of sexual partners in the last 12 months into a categorical variable (0, 1, 2+). We considered respondents to be consistent condom users if they were currently in a sexual relationship and in the last 12 months, they always used condoms with their primary and (if applicable) secondary partner; using this same schema, we also assessed condom use at last sex. For Zambia only, the condom use variables were only asked of respondents who were currently sexually active with their primary and (if applicable) secondary partner.

Using a binary (yes vs no) variable, we identified respondents as transactional sex engagers if they had engaged in sex with a stranger or casual partner to receive cash or kind; somewhere to stay; support or money for their children or family; drugs, food, cosmetics, clothes, a cellphone, airtime, transportation; somewhere to sleep for the night; financial support for bills or school fees; or anything else the they could not afford by themselves in the past 12 months.[41–43]

We measured intimate partner violence (IPV) among all respondents who reported having a sexual partner in the last 12 months and classified respondents as having experienced sexual IPV if they disclosed that in the last 12 months, they had sex or performed other sexual acts against their will because a current or previous boyfriend or partner used physical coercion, threats and intimation, or force. We coded respondents as having experienced physical IPV if they had been slapped or had something thrown at them; pushed or shoved; hit with a fist or another dangerous object; kicked, dragged, beaten, choked, or burnt; or threatened with a gun, knife or other weapon by a current or previous boyfriend or partner in the last 12 months.[11]

Lastly, questions on non-partner sexual violence were asked of all AGYW. We classified respondents as having experienced non-partner sexual violence if they disclosed that in the last 12 months, a person other than their boyfriend or partner perpetrated unwanted sex by using persuasion or force—whether successful or unsuccessful; forced sex while they were under the influence of drugs or alcohol and too impaired to refuse; or forced sex by two or more men—when the respondent was either sober or under the influence of drugs or alcohol.[11]

## Data analysis

We conducted bivariate $\chi^2$ analyses to examine the change in outcome measures between data collection rounds. To account for intraindividual correlation, we used generalised estimating equations (with a Poisson distribution) to examine temporal changes, control for characteristics (study site and marital, schooling and orphanhood status at round 1), and report measures of association (adjusted incidence rate ratio (IRR) (95% CI)). Adjusted models were run with and without adjusting for interruptions in programme participation variable to assess if level of programme exposure influenced outcomes. To achieve model convergence, we removed marital status from the

model examining transactional sex among Zambian AG. We ran all analyses using Stata (V.15.0).

## Patients and public involvement

This study actively engaged the study community and programme IP in the research process. Approval for the study was sought from community leaders and representatives in each setting during study planning. Research questions and tools were vetted with programme IP prior to administration. Postdata collection, dissemination meetings were conducted with programme IP and community leaders to share findings and implications for HIV prevention and other programming for AGYW in the study communities. Short study results briefs and presentation slide decks were developed and shared with programme IP and other collaborators for dissemination of research results in each setting with key leaders and decision-makers.

## RESULTS

### Cohorts' characteristics

Table 1 presents the sociodemographic characteristics of the cohorts in Kenya, Malawi and Zambia. Overall, the mean baseline age was between 17 (Kenya, Zambia) and 18 (Malawi) for AG and about 22 for YW (all countries). At round 1, most (>50%) Kenyan and Zambian respondents were unmarried. Conversely, marriage was common among Malawian AG (62%) and YW (89%) at round 1. The majority of AG in Kenya and Zambia were in school (>60%) at round 1, and over 50% of Zambian YW were still in school at round 2. Across all sociodemographic characteristics, there were modest shifts over time, as only a few age groups in certain countries had significant increases in out-of-school (Kenyan AG, Zambian AG and YW) and ever-married (Malawian AG and YW) status.

Respondents had nearly universal exposure to DREAMS' primary social asset building intervention: safe space groups, and exposure to HTS was high among Kenyan AG (96.1%) and YW (91.4%) but lower among AG and YW in Malawi and Zambia. With the exception of Kenya, where approximately ≥50% of AG and YW were exposed to educational support or economic support, exposure to other health services, educational support and economic support was low (eg, no YW in Malawi and 17% in Zambia had received some economic support).

On average, 75% of respondents experienced some form of programme interruption, with just under two in three reporting many (ie, ≥3) interruptions that lasted less than a week or more than a month. In Zambia, for instance, 75% of AG noted there were times when they were not able to participate in DREAMS with 29% reporting at least five interruptions lasting for a month or more. The main reasons for periods of non-participation included being away at school (Kenyan AG (58.9%)), sickness (Malawian AG (51.5%) and YW (47.1%)) and being away from the community (Kenyan YW (32.9%) and Zambian AG (29.0%) and YW (37.4%)).

**Table 1** Characteristics of cohorts in Kenya, Malawi and Zambia

| | Kenya | | | | Malawi | | | | Zambia | | | |
|---|---|---|---|---|---|---|---|---|---|---|---|---|
| | 15–19 n=389 | | 20–24 n=347 | | 15–19 n=371 | | 20–24 n=883 | | 15–19 n=487 | | 20–24 n=398 | |
| Characteristics | Round 1 | Round 2 | Round 1 | Round 2 | Round 1 | Round 2 | Round 1 | Round 2 | Round 1 | Round 2 | Round 1 | Round 2 |
| Mean age (SD) | 16.6 (1.3) | 18.3 (1.8) | 21.6 (1.4) | 22.6 (1.9) | 18.0 (1.2) | 19.3 (1.3) | 21.9 (1.4) | 23.1 (1.6) | 16.8 (1.4) | 18.1 (1.6) | 21.6 (1.3) | 22.5 (1.4) |
| Marital status | | | | | *** | | *** | | | | | |
| Ever married | 4.4 | 6.4 | 44.4 | 49.6 | 62.3 | 74.4 | 88.9 | 93.7 | 2.5 | 3.1 | 8.5 | 11.8 |
| Never married | 95.6 | 93.6 | 55.6 | 50.4 | 37.7 | 25.6 | 11.1 | 6.3 | 97.5 | 96.9 | 91.5 | 88.2 |
| Current schooling status | ** | | | | | | | | *** | | | ** |
| In school | 84.3 | 75.3 | 32.9 | 27.1 | – | – | – | – | 77.6 | 60.6 | 61.6 | 52.3 |
| Out of school | 15.7 | 24.7 | 67.1 | 72.9 | – | – | – | – | 22.4 | 39.4 | 38.4 | 47.7 |
| Orphanhood | | | | | | | | | | | | |
| Both parents alive | 62.5 | 58.1 | 42.4 | 38.9 | 70.9 | 69.5 | 61.4 | 60.8 | 64.9 | 62.6 | 55.5 | 54.3 |
| Lost at least one parent | 37.5 | 41.9 | 57.6 | 61.1 | 29.1 | 30.5 | 38.6 | 39.2 | 35.1 | 37.4 | 44.5 | 45.7 |
| Programme participation/exposure | | | | | | | | | | | | |
| Social asset building | – | 91.3 | – | 92.2 | – | 96.5 | – | 97.3 | – | 90.4 | – | 91.5 |
| Offered HTS | – | 96.1 | – | 91.4 | – | 62.0 | – | 65.1 | – | 80.9 | – | 73.9 |
| Offered other health service | – | 40.2 | – | 44.4 | – | 29.1 | – | 30.5 | – | 36.2 | – | 34.4 |
| Educational support | – | 66.8 | – | 57.6 | – | 1.4 | – | 1.1 | – | 46.4 | – | 21.9 |
| Economic support | – | 49.6 | – | 57.6 | – | 1.0 | – | 0.0 | – | 25.1 | – | 16.6 |
| Interruptions | | | | | | | | | | | | |
| None | – | 27.5 | – | 26.5 | – | 21.0 | – | 24.0 | – | 25.1 | – | 29.4 |
| <3 interruptions of ≤1 week | – | 6.9 | – | 6.3 | – | 9.7 | – | 7.1 | – | 5.5 | – | 6.5 |
| <3 interruptions of ≥1 month | – | 7.7 | – | 4.6 | – | 3.2 | – | 3.4 | – | 3.5 | – | 3.3 |
| ≥3 interruptions of ≤1 week | – | 23.1 | – | 28.8 | – | 22.1 | – | 28.0 | – | 37.0 | – | 30.4 |
| ≥3 interruptions of ≥1 month | – | 34.7 | – | 33.7 | – | 43.9 | – | 37.5 | – | 28.9 | – | 30.4 |

Other health services includes condoms, pre-exposure prophylaxis (where available), postviolence care, contraception, STI testing, pregnancy consultation.
Educational support includes money for fees, uniforms, transport or other help with schooling expenses (Kenya, Zambia) or back to school support (Malawi).
Economic support includes cash transfer (Kenya, Zambia) and food support (Malawi).
Programme participation/exposure and interruptions measured only at round 2.
Malawi's sample only included out-of-school participants; hence, current schooling status is not reported for Malawi's cohorts.
***p≤0.01; ***p≤0.001 for marital, schooling and orphanhood status.
HTS, HIV testing services; STI, sexually transmitted infection.

**Table 2** Temporal shifts in behavioural, biological, and experiential HIV risk factors among AGYW in Kenya

| | Aged 15–19 n=389 | | | | Aged 20–24 n=347 | | | |
|---|---|---|---|---|---|---|---|---|
| | Round 1 | Round 2 | P value | aIRR (95% CI) | Round 1 | Round 2 | P value | aIRR (95% CI) |
| **HIV service use** | | | | | | | | |
| HIV testing in last year | 81.5 | 96.4 | **<0.001** | **1.18 (1.02 to 1.37)** | 96.3 | 97.1 | 0.525 | 1.01 (0.88 to 1.15) |
| **Sexual behaviour** | | | | | | | | |
| STI symptoms | 14.4 | 14.9 | 0.839 | 1.04 (0.72 to 1.49) | 23.1 | 21.0 | 0.522 | 0.91 (0.68 to 1.22) |
| No of sex partners | | | 0.531 | 1.00 (0.81 to 1.23) | | | 0.259 | 1.00 (0.87 to 1.15) |
| 0 | 11.1 | 8.7 | | | 5.1 | 2.8 | | |
| 1 | 73.0 | 78.6 | | | 79.8 | 83.8 | | |
| 2+ | 15.9 | 12.7 | | | 15.1 | 13.4 | | |
| Consistent condom use | 52.5 | 45.8 | 0.395 | 0.78 (0.52 to 1.14) | 30.7 | 19.0 | **0.004** | **0.57 (0.40 to 0.81)** |
| Condom use at last sex | 74.6 | 64.4 | 0.172 | 0.79 (0.54 to 1.15) | 46.3 | 38.0 | 0.071 | 0.77 (0.59 to 1.01) |
| Transactional sex | 9.5 | 8.1 | 0.664 | 0.91 (0.49 to 1.68) | 5.7 | 10.0 | **0.044** | **1.73 (1.04 to 2.89)** |
| **Violence perpetrated by an intimate partner** | | | | | | | | |
| Sexual | 19.8 | 6.4 | **<0.001** | **0.32 (0.18 to 0.56)** | 20.2 | 10.5 | **0.001** | **0.51 (0.34 to 0.75)** |
| Physical | 15.2 | 14.7 | 0.866 | 0.96 (0.61 to 1.50) | 31.1 | 25.0 | 0.099 | 0.78 (0.60 to 1.02) |
| **Violence perpetrated by a non-partner** | | | | | | | | |
| Sexual | 21.1 | 19.5 | 0.593 | 0.93 (0.69 to 1.25) | 30.5 | 13.5 | **<0.001** | **0.44 (0.32 to 0.61)** |

Bolded cells indicate p≤0.05.
aIRR adjusts for location and round 1 marital, schooling and orphanhood status.
AGYW, adolescent girls and young women; aIRR, adjusted incidence rate ratios; STI, sexually transmitted infection.

## Kenya: unadjusted and adjusted transitions in AG and YW's HIV risk factors

Table 2 displays the unadjusted and adjusted changes in HIV risk factors among Kenyan AG and YW. Among AG, HIV testing increased (81.5% to 96.4%, p<0.001, aIRR=1.18 (1.02 to 1.37)) and reports of sexual IPV decreased (19.9% to 6.4%, p<0.001, aIRR=0.32 (0.18–0.56)). YW reported a decrease in consistent condom use (31.0% to 19.0%, p=0.004, aIRR=0.57 (0.40 to 0.81)), sexual IPV (20.2% to 10.5%, p=0.001, aIRR=0.51 (0.34–0.75)) and non-partner sexual violence (30.5% to 13.5%, p<0.001, aIRR=0.44 (0.32–0.61)) and an increase in transactional sex (5.7% to 10.0%, p=0.044, aIRR=1.73 (1.04 to 2.89)).

## Malawi: unadjusted and adjusted changes in AG and YW's HIV risk factors

Table 3 contains the unadjusted and adjusted shifts in HIV risk factors among Malawian AG and YW. AG reported more HIV testing (85.4% to 94.5%, p<0.001); they also reported fewer sex partners (2+sex partners: 9.4% to 1.8%, p<0.001) and experiences of sexual violence from partners (16.9% to 5.9%, p<0.001, aIRR=0.35 (0.21–0.57)) and non-partners (10.0% to 4.3%, p=0.003, aIRR=0.43 (0.25 to 0.74)). However, the associations for HIV testing and number of sexual partners became null in the multivariable models. Among YW, increases in HIV testing (82.5% to 91.7%, p<0.001, aIRR=1.11 (1.01 to 1.22)) were coupled with decreases in STI symptoms (31.8% to 22.1%, p<0.001, aIRR=0.69 (0.60 to 0.81)), number of

sexual partners (2+ sex partners: 6.2% to 3.2%, p<0.001) and experiences of sexual IPV (16.4% to 4.5%, p<0.001, aIRR=0.27 (0.20 to 0.38)), physical IPV (17.2% to 11.8%, p=0.001, IRR=0.68 (0.55 to 0.85)) and non-partner sexual violence (8.5% to 1.8%, p<0.001, aIRR=0.21 (0.13 to 0.36)). In the multivariable models, the association for number of sexual partners became null.

## Zambia: unadjusted and adjusted shifts in AG and YW's HIV risk factors

Table 4 presents the unadjusted and adjusted transitions in HIV risk factors among Zambian AG and YW. For AG, HIV testing increased over time (47.4% to 74.1%, p<0.001, aIRR=1.56 (1.34 to 1.82)). YW reported more HIV testing (67.6% to 76.1%, p=0.007) and transactional sex (2.4% to 5.8%, p=0.049) and fewer experiences of non-partner sexual violence (23.4% to 14.1%, p=0.001, aIRR=0.60 (0.44 to 0.82)). Increases in HIV testing and transactional sex exhibited a trend towards significance in the adjusted models.

## DISCUSSION

We present one of the first multicountry analyses examining temporal changes in behaviours and experiences associated with HIV acquisition among cohorts of AG and YW enrolled in DREAMS in Kenya, Malawi and Zambia. Our analysis uses prospective data and presents stratified analyses by age group. Our findings illuminate heterogeneous shifts in HIV-related risk factors between AG and

**Table 3** Longitudinal changes in behavioural, biological and experiential HIV risk factors among Malawian AGYW

| | Aged 15–19 n=371 | | | | Aged 20–24 n=883 | | | |
|---|---|---|---|---|---|---|---|---|
| | Round 1 | Round 2 | P value | aIRR (95% CI) | Round 1 | Round 2 | P value | aIRR (95% CI) |
| HIV service use | | | | | | | | |
| HIV testing in last year | 85.4 | 94.5 | **<0.001** | 1.11 (0.95 to 1.29) | 82.5 | 91.7 | **<0.001** | **1.11 (1.01 to 1.22)** |
| Sexual behaviour | | | | | | | | |
| STI symptoms | 25.6 | 22.4 | 0.302 | 0.87 (0.67 to 1.14) | 31.8 | 22.1 | **<0.001** | **0.69 (0.60 to 0.81)** |
| No of sex partners | | | **<0.001** | 1.00 (0.86 to 1.17) | | | **<0.001** | 1.07 (0.98 to 1.18) |
| 0 | 20.4 | 12.3 | | | 23.5 | 14.3 | | |
| 1 | 70.2 | 86.0 | | | 70.3 | 82.5 | | |
| 2+ | 9.4 | 1.8 | | | 6.2 | 3.2 | | |
| Consistent condom use | 11.8 | 6.8 | 0.056 | 0.56 (0.31 to 1.01) | 5.0 | 5.4 | 0.743 | 0.89 (0.57 to 1.38) |
| Condom use at last sex | 20.2 | 20.3 | 0.967 | 0.98 (0.66 to 1.43) | 14.0 | 15.7 | 0.395 | 1.01 (0.80 to 1.27) |
| Transactional sex | 2.4 | 3.5 | 0.412 | 1.40 (0.62 to 3.17) | 3.9 | 3.0 | 0.281 | 0.76 (0.48 to 1.22) |
| Violence perpetrated by an intimate partner | | | | | | | | |
| Sexual | 16.9 | 5.9 | **<0.001** | **0.35 (0.21 to 0.57)** | 16.4 | 4.5 | **<0.001** | **0.27 (0.20 to 0.38)** |
| Physical | 13.9 | 11.2 | 0.29 | 0.80 (0.55 to 1.17) | 17.2 | 11.8 | **0.001** | **0.68 (0.55 to 0.85)** |
| Violence perpetrated by a non-partner | | | | | | | | |
| Sexual | 10.0 | 4.3 | **0.003** | **0.43 (0.25 to 0.74)** | 8.5 | 1.8 | **<0.001** | **0.21 (0.13 to 0.36)** |

Bolded cells indicate p≤0.05.
aIRR adjusts for location and round 1 marital and orphanhood status. We did not control for schooling because out-of-school status was a participation requirement in Malawi.
AGYW, adolescent girls and young women; aIRR, adjusted incidence rate ratios; STI, sexually transmitted infection.

**Table 4** Changes over time in behavioural, biological and experiential HIV risk factors among AGYW in Zambia

| | Aged 15–19 n=487 | | | | Aged 20–24 n=398 | | | |
|---|---|---|---|---|---|---|---|---|
| | Round 1 | Round 2 | P value | aIRR (95% CI) | Round 1 | Round 2 | P value | aIRR (95% CI) |
| HIV service use | | | | | | | | |
| HIV testing in last year | 47.4 | 74.1 | **<0.001** | **1.56 (1.34 to 1.82)** | 67.6 | 76.1 | **0.007** | 1.13 (0.99 to 1.28) |
| Sexual behaviour | | | | | | | | |
| STI symptoms | 12.7 | 11.5 | 0.556 | 0.90 (0.65 to 1.25) | 15.8 | 15.8 | 1.000 | 1.00 (0.75 to 1.34) |
| No of sex partners | | | 0.111 | 0.92 (0.74 to 1.16) | | | 0.450 | 0.94 (0.81 to 1.10) |
| 0 | 7.6 | 16.1 | | | 15.0 | 17.0 | | |
| 1 | 85.9 | 76.5 | | | 72.1 | 73.3 | | |
| 2+ | 6.6 | 7.4 | | | 13.0 | 9.8 | | |
| Consistent condom use | 42.5 | 38.8 | 0.706 | 0.78 (0.45 to 1.35) | 42.3 | 33.3 | 0.167 | 0.72 (0.47 to 1.09) |
| Condom use at last sex | 60.0 | 55.2 | 0.629 | 0.87 (0.52 to 1.44) | 52.6 | 45.5 | 0.287 | 0.81 (0.57 to 1.16) |
| Transactional sex | 0.9 | 3.7 | 0.163 | 3.76 (0.45 to 31.67) | 2.4 | 5.8 | **0.049** | 2.50 (0.99 to 6.35) |
| Violence perpetrated by an Intimate partner | | | | | | | | |
| Sexual | 17.6 | 17.9 | 0.923 | 1.04 (0.73 to 1.47) | 18.4 | 14.4 | 0.144 | 0.79 (0.56 to 1.11) |
| Physical | 23.3 | 23.9 | 0.859 | 1.03 (0.77 to 1.36) | 15.6 | 12.1 | 0.172 | 0.75 (0.52 to 1.09) |
| Violence perpetrated by a non-partner | | | | | | | | |
| Sexual | 12.5 | 10.9 | 0.425 | 0.87 (0.61 to 1.23) | 23.4 | 14.1 | **0.001** | **0.60 (0.44 to 0.82)** |

Bolded cells indicate p≤0.05.
aIRR adjusts for location and round 1 marital, schooling, and orphanhood status
To achieve model convergence, the transactional sex model for AG does not include the respondent's marital status at round 1.
AGYW, adolescent girls and young women; aIRR, adjusted incidence rate ratios; STI, sexually transmitted infection.

YW, and across countries. Over time, AG who were part of DREAMS report increases in HIV testing (Kenya and Zambia) and decreases in sexual IPV (Kenya and Malawi) and non-partner sexual violence (Malawi). HIV testing increased among YW in Malawi and Zambia (borderline significance); Malawian YW also had fewer experiences of STIs. Additionally, YW report significant or marginal increases in transactional sex (Kenya and Zambia) and decreases in consistent condom use (Kenya). YW engaged in DREAMS also report lower sexual IPV (Kenya and Malawi), lower physical IPV (Malawi) and non-partner sexual violence (Kenya, Malawi and Zambia). Some emerging research in Uganda report that DREAMS programming contributed to the reduction of sexual risk behaviours; though, the effects differed by age group.[44] While the DREAMS programming models included considerations for participant age (eg, providing education support to AG, and livelihoods support to YW), our findings indicate that additional tailoring of comprehensive HIV prevention packages may be needed to address the risk heterogeneity of AG and YW across different contexts.[43]

Increases in HIV testing rates among Kenyan and Zambian AG and Malawian YW, including a notable increase among Zambian YW, are promising as HIV testing is the first step in the HIV care cascade and remains a key prevention strategy. Our data show that close to 75% (or greater) of AG and YW self-reported testing for HIV in the last year. In comparison, recent national averages of self-reported HIV testing are much lower (Kenya: 35% of AG, 64% of YW; Malawi: 28% of AG, 52% of YW; Zambia: 29% of AG, 53% of YW).[3 4 45] These results likely point to PEPFAR, local national HIV programmes and policies, and DREAMS' focus on expanded access to HTS to AGYW in the DREAMS study communities.[29] Though only observed within one group, it is also encouraging to see that Malawian YW reported fewer STI symptoms over time, as STIs increase AGs' and YW's biological vulnerability to HIV.[46 47] Within DREAMS, establishing, providing, and strengthening youth-friendly health services was a key part of the intervention package. AG and YW were primarily linked to these services through discussions and referrals provided in the safe space groups. This community-based platform may have been particularly helpful in bringing services to marginalised AGYW, who would otherwise be hard to reach with HIV and reproductive health programming.[48]

Decreases in experience of sexual IPV over time are also promising. Previous research supports the direct relationship between HIV acquisition and sexual IPV, primarily because it facilitates other risk behaviours and mechanisms (eg, decrease in self-efficacy and increases in multiple partnerships, transactional sex, substance abuse, negative mental health outcomes and biological vulnerability due to trauma).[10 11 49 50] Compared with data from nationally representative surveys among ever-married women aged 15–49, the baseline prevalence for sexual IPV among AGYW in our cohorts are high.[45 51]

The DREAMS programmes were conducted in high-risk communities with the most-at-risk AGYW, which may account for the higher experiences of sexual violence than the national average. Over time, we find that AG and YW in Kenya and Malawi reported significantly fewer incidents of sexual IPV over time. Since it is also perpetrated by a trusted individual, physical IPV has similar mediating pathways.[10 52] In, almost, all countries and age groups, physical IPV reduced, yet the reduction was only significant among Malawian YW and could be because most Malawian YW were ever married at round 1 and, therefore, had the greatest opportunity for positive change; they also had the highest mean age, which could have translated into greater agency and relationship power. Additional evidence is needed to support or refute these hypotheses. Zambian AG and YW reported no significant shifts related to either IPV outcome; this warrants further investigation as to how IPV is perceived among Zambian AGYW and how programmes are attempting to address it.

Sexual violence perpetrated by non-partners (assault and rape) is understudied; in our analysis, the baseline prevalence of non-partner sexual violence ranged from 10.0% to 21.1% among AG and 8.5% to 30.5% among YW. These totals are likely underestimated due to the topic's sensitive nature, which can trigger recollections of sadness and trauma; regardless, compared with other non-partner sexual violence metrics (9.1% prevalence among Johannesburg AG,[13] 11.9% (8.5%–15.3%) lifetime prevalence among African women,[52] these baseline totals are high. Other studies have found that surviving non-partner sexual violence can lead to a myriad of behaviours or outcomes that heighten HIV risk (eg, substance abuse, poorer sexual and reproductive health and mental health outcomes, and increased experience of STIs).[11 13 52] In our study, it is reassuring that we observed appreciable drops in non-partner sexual violence among DREAMS participants across all age groups and countries, which hopefully signifies community-wide shifts in accountability around violence and safer spaces for AGYW. DREAMS' multisectoral approach could explain some of the notable decrease in IPV and non-partner violence. Gender-inequitable social norms and patriarchal power structures, often, create environments where men are emboldened to perpetrate violence, both against their intimate and strangers/casual partners. To disrupt these norms, DREAMS programming made a concerted effort to create an enabling environment for AGYW by challenging existing power dynamics and addressing gender-based violence—both within the safe space groups and the broader programming community.[29–32] However, since all girls were exposed to anti-violence programming and our study does not include any data from men or other community members, this study cannot disentangle its effects from other activities. To strengthen our understanding, future research should consider collecting attitudes and behaviours related to non-partner violence from various ecological strata, and further examination is required to unpack the specific programming

components and pathways that may have contributed to these significant changes.

We do not find substantial shifts in other preventive behaviours (ie, increase in condom use, decrease in multiple partnerships). On average, consistent condom use and condom use at last sex decreased between survey rounds, with the greatest decrease occurring among Kenyan YW. While previous research shows that proper condom use is protective against HIV acquisition,[53 55] condom uptake among adolescent and youth populations is consistently suboptimal,[53 56–59] a pattern that is reinforced by our findings. A post hoc trend decomposition analysis suggests that condom use behaviours may have shifted for YW as they entered martial partnerships (results not shown). Previous literature suggests that low condom use among AG and YW could be a product of failed negotiations fueled by low self-efficacy, relationship power and gender-inequitable norms and practices.[6 18 43 60 61] Alternatively, condom use could have decreased due to changes in contraceptive use—such as moving from condoms to hormonal contraceptives—or desires to have children with their spouse.[62–64] Multiple factors and motivations may be contributing to shifts in condom use among AG and YW, especially those in recent unions. For HIV prevention programming, it will be critical to consider approaches to ensure that AGYW and their partners employ strategies (eg, couple HIV testing and counselling, safer conception practices) to mitigate HIV risk as they transition in and out of relationships and have children.

Among Kenyan and Zambian YW, we found significant or trending increases of transactional sex over time. Previous research shows that transactional sex can increase women's HIV risk.[19 20 41 65 66] To avert poverty, homelessness and food insecurity, some may turn to transactional sex to survive or to obtain desired commodities.[17 18 67] AGYW's economic disenfranchisement is characterised by a constellation of factors—including gendered discrimination and lack of social and business capital[68]—that make it difficult to AGYW to access the economic resources or earn a living wage. While DREAMS provided some economic activities (eg, financial literacy and education) and strengthened access to some economic resources (eg, access to school bursary programmes or savings accounts), it is possible that broader partnerships or deeper investments are needed to make gainful employment/entrepreneurship possible for AGYW. Additionally, it is possible that the implementation of the socioeconomic interventions within DREAMS may have faltered. Research in South Africa on the DREAMS Partnership found that rollout of programme components that required multisectoral collaborations (especially those beyond the health sector) were complex and challenging in a community-based setting.[69] Other structural intervention programmes have reported reductions in economically driven sex and risky sexual practices (eg, condomless sex, multiple sex partners).[70–73] These positive shifts show that intersectional approaches can lead to meaningful change and should continue to be considered by future HIV prevention programmes for AGYW.[74–76]

While we found some promising results, this study is not without limitations. Across the three countries, 17%–25% of the respondents could not be reinterviewed at round 2; however, this level of attrition is consistent with other community-based safe space programmes.[77] While our original sample size calculations were estimated anticipating this lost to follow-up and the lost to-follow-up analysis (online supplemental table 1) shows similar distributions for most baseline characteristics and outcomes, this attrition could have introduced selection bias. To avoid this in future studies, robust retention techniques should be employed. In Zambia, we could only assess condom use among a specific subset of sexually active respondents, thus our findings should not be generalised to all sexually active AGYW. All the data were self-reported, which is prone to social desirability and recall bias. In our survey, we assessed level of programme engagement/continuity of programme exposure but adjusting for programme exposure did not change the regression results; additional attention may be needed for more robust measures on programme exposure. An additional challenge was the lack of a control group; since the DREAMS programme was implemented over time, it continued to enrol AGYW from nearby communities and catchment areas, making it hard to find an appropriate counterfactual group. It also means our findings could be prone to confounding by secular trends and the ageing of the cohorts in the study contexts. Subsequent analyses using modelling techniques using age as the timescale to create a control group could further help to reduce bias.

## CONCLUSION

Recent large-scale HIV prevention efforts are currently underway to reduce the disproportionate burden of HIV among AGYW in SSA. Our longitudinal analysis assessed temporal change in key behaviours and experiences associated with HIV acquisition among Kenyan, Malawian and Zambian AGYW cohorts nested within DREAMS, a multisectoral HIV prevention programme. These analyses found a positive effect on some HIV risk factors, such as HIV testing, non-partner sexual violence, and physical and sexual IPV, but variable improvements in sexual risk behaviours. Our findings contribute to the growing body of evidence that is assessing the impact of community-based multisectoral HIV prevention programming for AGYW.

**Author affiliations**
[1]Population Council, Washington, District of Columbia, USA
[2]Population Council, New York, New York, USA
[3]Population Council India, New Delhi, Delhi, India
[4]Population Council Kenya, Nairobi, Kenya
[5]Centre for Reproductive Health, University of Malawi College of Medicine, Blantyre, Southern Region, Malawi
[6]Independent Consultant, Lusaka, Zambia

[7]ViiV Healthcare US, Research Triangle Park, North Carolina, USA

**Correction notice** This article has been corrected since it was first published.

**Acknowledgements** The authors would like to acknowledge the implementing partners in Kenya (Aphia Plus/PATH, I Choose Life, Make Me Smile), Malawi (OneCommunity) and Zambia (PACT, CMMB, Marie Stopes Zambia), and our research team coordinators and data collectors for their support during the data collection. We would like to acknowledge support from Population Council colleagues who provided support in the preparation of this manuscript, and our deepest gratitude goes to the study participants. This work would not have been possible without the active support of the PEPFAR DREAMS teams in Kenya, Malawi and Zambia.

**Contributors** SM, NP and JP conceptualised the study. SM, NP, CJH, SKP and BM devised and implemented the analysis plan. EC, VM, WC, JO and MM supported the study design and led the data collection. SM and CJH led the manuscript writing. All authors reviewed and revised the manuscript. SM acts as guarantor for the manuscript.

**Funding** Funding support for the Kenya and Zambia studies was provided by the Bill & Melinda Gates Foundation (OPP1136778, DREAMS Implementation Science Research). Funding for the Malawi study was provided by the generous support of the American people through the United States President's Emergency Plan for AIDS Relief (PEPFAR) and the US Agency for International Development (USAID) under Project SOAR (Cooperative Agreement AID-OAA-A-14-00060).

**Disclaimer** The contents of this manuscript are the sole responsibility of the authors and do not necessarily reflect the views of PEPFAR, USAID, or the United States Government.

**Competing interests** NP works full-time for ViiV Healthcare. ViiV Healthcare did not fund the study. The contents of the manuscript do not reflect the views of ViiV Healthcare. SM, CJH, SKP, JO, EC, VM, WC, MM, BM and JP declare no conflicts of interest.

**Patient consent for publication** Not applicable.

**Ethics approval** This study involves human participants and was approved by at each survey round, we collected written informed consent from adult participants (those above the age of consent in each country) and emancipated minors (when applicable). If the participant was considered a minor, we obtained parental consent and participant assent. The protocols for these studies were reviewed and approved by Population Council's Institutional Review Board (protocol # 744 and 745); Kenya's Kenyatta National Hospital/University of Nairobi Ethics (P384/05/2016) and Research Committee and National Commission for Science, Technology and Innovation; Malawi's College of Medicine Research and Ethics Committee at the University of Malawi (PP.01/17/2095); and Zambia's ERES Converge Institutional Review Board (2016- May-016) and National Health Research Authority. Participants were compensated Ksh300 (~US$3), MK1500 (~ US$2) or ZK50 (~US$5) for their time, per local ethical guidelines. Participation in this study did not influence access to or participation in any of the DREAMS interventions. Participants gave informed consent to participate in the study before taking part.

**Provenance and peer review** Not commissioned; externally peer reviewed.

**Data availability statement** Data are available in a public, open access repository. Data are available on reasonable request. All the underlying data relevant to the study are fully available on Dataverse.

**ORCID iD**
Craig J Heck http://orcid.org/0000-0002-8138-6640

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
