## [Reviewer comments · BMJ Open]

ARTICLE DETAILS

TITLE (PROVISIONAL)	Temporal shifts in HIV-related risk factors among cohorts of adolescent girls and young women enrolled in DREAMS programming: Evidence from Kenya, Malawi, and Zambia
AUTHORS	Mathur, Sanyukta; Heck, Craig; Kishor Patel, Sangram; Okal, Jerry; Chipeta, Effie; mwapasa, victor; Chimwaza, Wanangwa; Musheke, Maurice; Mahapatra, Bidhubhusan; Pulerwitz, Julie; Pilgrim, Nanlesta

VERSION 1 – REVIEW

REVIEWER	Barnhart, Dale Harvard University T H Chan School of Public Health, Department of Epidemiology
REVIEW RETURNED	12-Feb-2021

GENERAL COMMENTS	I have a very long list of statistical recommendations, but I do want to congratulate you on strong data collection on this important topic and wish you the best of luck in revisions
--

REVIEWER	McKinnon, L University of Manitoba
REVIEW RETURNED	15-Mar-2021

GENERAL COMMENTS	How was enrolment done? Sites are described but not inclusion criteria, how AGYW were informed about the project, etc. "Who" these populations are is really critical to understanding the outcomes, and this is lacking in this draft. The results are often vague and a bit confusing- perhaps there could be a combined analysis across sites, with differences pointed out when relevant. Otherwise, why combine all cohorts into the same paper? This could be done statistically. The intervention isn't quite clear. Some of it can be inferred throughout the paper, but a clearer definition would be helpful. Also, a lot of work goes in to describing intervention exposure, yet this does not appear to feature in explaining increases/decreases observed? Lack of comparison group makes it hard to know what the intervention contributed to any changes. While it is clear that this program was to be made available to anyone interested, there could have been some attempt – like a step-wedge design, that could have tried to answer what the intervention did. Becoming more sexually active over time does not seem like a useful endpoint, as it is fairly obvious this will occur (given the age of participants) if the study is prospective. That there is an increase in
--

	sexual debut and marriage, and individuals leaving school does not seem that surprising – in a way, the study is tracking life events, more than assessing an intervention? The decomposition analysis to determine whether sociodemographic shift influenced temporal changes is not particularly well explained. The results presented are very difficult to interpret as no statistics are provided. Supplemental table 1 is not useful. In those not followed up, is there a difference in baseline characteristics from those who were followed? Figure 1 and 2 should have confidence intervals or some other measure of uncertainty on pre and post frequencies of each outcome measure. Lines 433-37: Condom use decreasing during marriage may also decrease due to fertility desires, or use of a longer lasting form of birth control, rather than any of the power dynamics that might be more critical in shorter term more transient relationships. Increase in transactional sex despite the economic aspects of DREAMS is a bit difficult to interpret. Could this be discussed further?
--	--

VERSION 1 – AUTHOR RESPONSE

Reviewer 1:

I have a very long list of statistical recommendations, but I do want to congratulate you on strong data collection on this important topic and wish you the best of luck in revisions!
Thank you very much, we appreciate your support and careful review.

The subject is very timely and important and the data collection process seems really strong. Thank you.

However, the analysis plan is not well developed. I would recommend the following suggestions. Of course, these could change the results and their interpretation substantially, but I think they could really improve the interpretation of your data. We have attempted to address this comment and further clarify our analytical approach and results.

Comment	Reply
Major Comments	
Present a full table one that details the baseline characteristics of individuals who did and did not respond to the follow-up survey. Include the baseline levels of you outcomes of interest, including HIV service use, sexual behavior, and violence	Excellent point! We have added a supplemental table presenting baseline differences between those followed and those lost to follow-up.

outcomes in this table.	
Rather than make within-person pre-post comparisons where calendar year is the time scale, an alternative model technique would be to conduct a longitudinal model where age is the timescale. This approach would create a control group that is less confounded by the aging of the overall cohort. For example, the 17-year-old from baseline could be the control for a 16-year-old at baseline, who becomes 17 at the time of the follow-up survey. This approach would still be biased by secular trends, and I think it is okay if you choose to start with the described pre-post analysis for an initial paper, but would consider this as a promising next step for this or future papers.	Thank you for this suggestion and description of another longitudinal approach. We agree that this pre-post paper is just the first step in understanding the temporal effects of the DREAMS program among AGYW in Kenya, Malawi, and Zambia. For subsequent ideas and analyses, we will consider using age as the timescale to control for the confounding nature of aging. We have included this as a potential future research idea in the paper.
For your regression analyses, I would recommend that you pool your data across all three countries for your initial set of models. Many of the “differences” across the countries that you are finding difficult to interpret really reflect that the point estimate from one country is statistically significant and the other is not, even though the confidence intervals for both countries overlap substantially. That means that these differences could largely reflect issues with random variation and statistical power rather than true between-country differences. In table 3, violence against young women is a good example of this.	Thank you. We carefully considered the option to pool data across the three countries, but there are two reasons we decided not to revise the analytical approach. First, our goal was to present the country-specific findings (examining pre-post shifts on key HIV-related outcomes by age group) rather than analyze differences across countries. This is because the context to HIV is different across these settings and the nuance will be lost if we conduct a pooled country analysis. Second, we have also found that stakeholders and decision-makers are keen on country-specific information to feed into program considerations, thus we have retained the country-specific analysis here. Upon reviewing the manuscript, we did realize that the way we had presented the results (by age-group), inherently, gave the perception of inter-country comparisons. We have re-structured the manuscript so that the findings are presented by country instead. This is change is reflected in the text and tables.
If, after looking at aim 2, you are still interested in assessing for effect differences across countries, do a second set of models that include an interaction term in your	Thanks, again, for this great analytical comment. As above, this pre-post analysis is but the first step in assessing temporal shifts among AGYW. As we plan out subsequent analyses to assess

model and only interpret differences between countries if there is statistical evidence of an interaction. It could be very reasonable an appropriate to still have country-specific tables in the Appendix, especially if that were interesting to the individual programs.	inter-country differences in effects, we will absolutely consider this approach.
You cannot use this repeated measures Poisson model to assess the impact of your intervention on sexual debut – within an individual, sexual debut happens only once, therefor the “risk” can only increase over time. For this particular outcome, rather than look at the variable by pre/post, you could look at this by age (so someone who contributed data to the 16-year-old category in the first round of the survey could contribute data to the 17-year old category in the second round of the survey). This point is similar to point 2.	After careful consideration, we decided to remove the sexual debut outcome from the analysis because sexual debut is a logical behavior as AGYW get older. This edit has been made throughout the manuscript.
Do not adjust for round two variables, including continuity of program exposure and changes in education, marital, or orphanhood status, in your primary Poisson models for tables 2 and 3. This could be thought of adjusting for mediators and will bias your estimates for the overall effect of the intervention effect. If you are interested in doing a mediation analysis, there are particular methods for that, but reporting on the total non-mediated effect would be the first step.	In the multivariable Poisson models, we only included round one variables for education, marital, and orphanhood status. Continuity of program exposure is the only one variable that was measure at Round 2. We think it is important to adjust for this to make sure the changes are not because of variations in exposure to the program. We realize that the footnotes for the multivariable tables were not explicit about this point, so we added this information to the new tables (i.e., Tables 2, 3, and 4).
If you are interested in doing an “as treated” analysis to assess the effect size if everyone had had services without interruption, I would encourage you to incorporate inverse probability weights to your analysis.	Thank you for this innovative suggestion. We are not doing an “as treated” analysis in this paper. We are also limited in our ability to do conduct intent to treat analysis for our sample as we do not have outcome data for individuals who were lost to follow-up. However, we will consider this recommendation for our ongoing and future analyses.
You could also consider using inverse probability weights to adjust the analysis for loss to follow-up.	Yes, one can consider using IP weight. However, it is unclear if this approach would add significant value to this analysis. Our original sample sizes were calculated anticipating loss of attrition, and our studies are sufficiently powered to detect change over time. Therefore, we have decided not to use IP weight to adjust for loss to follow-up.

Regarding your decomposition analysis, it would be helpful if you could provide more details on the particular methods you are using, either in the methods or through citations. I think that what you are describing is what I would consider to be a mediation analysis. If it is just a difference in terminology that is okay, but it is hard to be certain that the methods are appropriate without additional details.	Thank you for this comment. We did not conduct a mediation analysis. We had conducted a post-hoc trend decomposition to assess if shifts in key socio-demographic characteristics contributed to the change over time in the study outcomes. To further simplify and focus the results presentation and our interpretations, we have removed the decomposition analysis and description in the paper, and only note it briefly in the discussion section.
Minor comments	
Abstract & similar	
Line 78: lack of a comparison group leaves you open to confounding by secular trends and the aging of the cohort. It does not have implications for generalizability.	We have updated the “Strengths and limitations of this study” section to reflect this point. We also added it to the limitations paragraph near the end of the manuscript.
Similarly, moderate loss-to-follow-up impedes study validity	Since the “Strengths and limitations of this study” section only allows 5 bullets, we added this point to the limitations paragraph in the manuscript.
Methods	
Could you explain the safe spaces idea a little bit more? Do you just mean a designated space and time to socialize? It may help readers understand a little bit better if you could provide a few examples (after school meeting? church meetings on weekends?)	We elaborated on the safe spaces in the “Program description” section.
Line 147: Could you list the specific primary package items?	We added additional language to describe the DREAMS core package of interventions.
Line 157-159: There seems to be missing text regarding the size of the beneficiary population in Malawi.	We have added the size of the beneficiary population in Malawi.
Line 194: Generally, the language around adjusting for schooling, except in Malawi, is confusing. My understanding is that, in Malawi, only out of school youth were eligible to participate in the program? I would explain this once in eligibility, once in modeling (e.g. in. vs. out of schooling was excluded in all Malawi-specific models since being out of school was a requirement for participation) and as a footnote in your tables.	Thank you for this excellent suggestion. We streamlined the text to reflect this suggestion.

In your data analysis section, it if unclear is you accounted for within-individual correlation, within group/site correlation, or both. A minimum, you need to account for within-individual correlation (which I think is what you are doing).	For the analysis, we accounted for intra-individual correlation by using generalized estimating equations and setting the cluster variable as the individual. We included the site variable as a covariate in each country's multivariable model to control for potential site-level differences. To better reflect this, we change "within-group correlation" to "intra-individual correlation" in the Data Analysis section.
Discussion	
When interpreting the decrease in physical and sexual violence, think about how loss to follow-up could affect that interpretation. These experiences could strongly motivate people to leave the community and drop out of your study. How high would incident assault have to be in the group that was lost to follow-up to erase the observed reduction in violence among the group that responded to the survey?	Thanks for this important point, the new supplemental table presents a comparison of baseline characteristics of individuals who did and did not respond to the follow-up survey. The data show that respondents and non-respondents were similar on baseline reports of most outcomes, there were no systematic patterns of differences. Experiences of sexual violence from intimate partners was significantly different/higher for 20-24 year olds in Malawi and 15-19 year olds in Zambia between respondents who were followed and those we were unable to reach, but these differences were not present for 15-19 year olds in Malawi or 20-24 year olds in Zambia or present in Kenya. Due to the lack of a systematic pattern, we do not believe that violence experience is the reason that study individuals were lost to follow-up. Additionally, as we note in the methods in the "Study Population" section, loss to follow-up most often occurred due to extended/permanent out migration from the study community.
In your limitations section, I would emphasize that the lost to follow-up could lead to selection bias, not just reduced power.	We added this to the limitations section, along with the possibility that it could have impact internal validity.
Grammar & General readability	
Excessive commas on lines, 92-93, 104, 109, reduce readability.	We reduced the commas on these lines to improve readability.
Similarly, the multiple parenthesis and bracket ([like this]) is very distracting. It would be better to reword you manuscript to minimize these parenthetical statements	Thank you for this feedback. Outside of reporting confidence intervals, we removed a large majority of them from our manuscript by re-writing and re-structuring the text.
Weird parenthetic commas are not technically incorrect, but are difficult to read. These sentences could be re-worded to improve readability 114-115;	Thank you. We edited these lines to help improve readability.
Line 13: Change was to were	Thank you, we have made this change on line 177.

Comment	Reply
How was enrolment done? Sites are described but not inclusion criteria, how AGYW were informed about the project, etc. “Who” these populations are is really critical to understanding the outcomes, and this is lacking in this draft.	Thank you for this comment, the “Study population” section of the methods describes the study enrolment and inclusion criteria.
The results are often vague and a bit confusing- perhaps there could be a combined analysis across sites, with differences pointed out when relevant. Otherwise, why combine all cohorts into the same paper? This could be done statistically.	Thank you for this recommendation. Apologies for our lack of clarity in the manuscript: our goal was to present the country-specific findings rather than analyze differences across countries. Upon reviewing the manuscript, we realize that presenting the findings by age-group, inherently, gives the perception of inter-country comparisons. Thus, we decided to re-structure the manuscript so that the findings are presented by country rather than age-group. This change is reflected in the text, figures, and tables.
The intervention isn’t quite clear. Some of it can be inferred throughout the paper, but a clearer definition would be helpful. Also, a lot of work goes in to describing intervention exposure, yet this does not appear to feature in explaining increases/decreases observed? Lack of comparison group makes it hard to know what the intervention contributed to any changes. While it is clear that this program was made available to anyone interested, there could have been some attempt – like a stepped-wedge design, that could have tried to answer what the intervention did.	Thank you for these comments. We have provided additional clarity on the DREAMS core intervention package in the “Program description” section of the methods. In this paper we are assessing temporal changes among adolescent girls and young women before and after exposure to the DREAMS core package of interventions. The package is delivered holistically, and the discussion reflects on some of potential influence that the package of interventions may have had on the outcomes assessed. We agree that a lack of comparison group is a limitation. Our limitations section notes this and its implications for the study findings. A stepped-wedge design is a wonderful suggestion for a future study design. Unfortunately, this package of interventions was implemented at-scale and provided to all AGYW included in the study. Given how the data were measured and captured, we cannot employ such a design retrospectively with the current dataset. However, we will keep a stepped-wedge design in mind for future studies.
Becoming more sexually over time does not seem like a useful endpoint, as it is fairly obvious this will occur (given the age of participants) if the study is prospective. That	Thank you for this comment. After careful consideration, we decided to remove the sexual debut outcome from the analysis because—as

there is an increase in sexual debut and marriage, and individuals leaving school does not seem that surprising – in a way, the study is tracking life events, more than assessing an intervention?	you note—sexual debut is a logical behavior as AGYW get older.
The decomposition analysis to determine whether sociodemographic shift influenced temporal changes is not particularly well explained. The results presented are very difficult to interpret as no statistics are provided. Supplemental table 1 is not useful.	Thank you for this comment, we agree. We conducted the decomposition analysis as a post-hoc analysis. Therefore, in reflecting on your comment and to further simplify and focus the paper, we have removed this emphasis from the paper. We now describe it briefly in the discussion section.
In those not followed up, is there a difference in baseline characteristics from those who were followed?	We have added a supplemental table (Supplemental Table 1) that compares the baseline characteristics of those who were followed vs. those who were not. Additionally, the “Study population” section of the methods presents information on comparison of baseline characteristics. For the most part, baseline characteristics were similar among those who were followed and those who were not; there were no systematic patterns of difference.
Figure 1 and 2 should have confidence intervals or some other measure of uncertainty on pre and post frequencies of each outcome measure.	We decided to remove Figures 1 and 2 from the manuscript. Instead, we report the proportions in Tables 2, 3, and 4.
Lines 433-37: Condom use decreasing during marriage may also decrease due to fertility desires, or use of a longer lasting form of birth control, rather than any of the power dynamics that might be more critical in shorter term more transient relationships.	Fantastic suggestion! In addition to the potential of differential power dynamics, we added text on the potential effect of fertility desires and contraceptives on condom use.
Increase in transactional sex despite the economic aspects of DREAMS is a bit difficult to interpret. Could this be discussed further?	Thank you for this comment, we have elaborated a bit more on this point (lines 451-455). While DREAMS did attempt to provide socio-economic interventions, the program implementation and the level of support may have been insufficient to fully reduce economic vulnerability among AGYW in these community contexts.

VERSION 2 – REVIEW

REVIEWER	Barnhart, Dale Harvard University T H Chan School of Public Health, Department of Epidemiology
REVIEW RETURNED	01-Sep-2021

GENERAL COMMENTS	This manuscript is clear and presents highly relevant information. Although I think including "program interruptions" in table 1 is extremely interesting and relevant as it speaks to the implementation process, my main critique of this manuscript is the inclusion of "program interruptions" in the adjusted models. It is difficult to conceptualize this variable as confounder that should be adjusted for and so it should not be included as a covariate. If you hypothesize that interruptions modify the effectiveness of DREAMS, which seems likely and plausible, that would best be explored using an interaction term, stratified analysis or controlled direct effect analysis, and would likely be the focus of a subsequent, in depth analysis. Given the general concordance between the unadjusted and adjusted results, it is unlikely that omitting this variable from the regression analysis would make many substantive changes to the paper.
--

REVIEWER	McKinnon, L University of Manitoba
REVIEW RETURNED	15-Sep-2021

GENERAL COMMENTS	all good!
-----------

VERSION 2 – AUTHOR RESPONSE

Reviewer 1 comment: Although I think including "program interruptions" in table 1 is extremely interesting and relevant as it speaks to the implementation process, my main critique of this manuscript is the inclusion of "program interruptions" in the adjusted models. It is difficult to conceptualize this variable as confounder that should be adjusted for and so it should not be included as a covariate. If you hypothesize that interruptions modify the effectiveness of DREAMS, which seems likely and plausible, that would best be explored using an interaction term, stratified analysis or controlled direct effect analysis, and would likely be the focus of a subsequent, in depth analysis. Given the general concordance between the unadjusted and adjusted results, it is unlikely that omitting this variable from the regression analysis would make many substantive changes to the paper.

Author response: Thank you for your careful review of our manuscript. Based on your input, we decided to remove program interruptions as a covariate in the multivariable models and, thus, only report program interruptions in Table 1. The revised manuscript presents the updated multivariable findings. As you predicted, the new findings are not substantively different from our original findings. We have made edits throughout the paper to reflect the new models. Per your suggestion, we will consider exploring program interruptions via stratification in a subsequent manuscript.

Reviewer 2 comment: all good!

Author response: Thank you for critical appraisal and input!